# Long Pentraxin 3 as a New Biomarker for Diagnosis of Hip and Knee Periprosthetic Joint Infections

**DOI:** 10.3390/jcm12031055

**Published:** 2023-01-29

**Authors:** Mattia Loppini, Marco Di Maio, Roberta Avigni, Roberto Leone, Antonio Inforzato, Guido Grappiolo, Alberto Mantovani, Barbara Bottazzi

**Affiliations:** 1IRCCS Humanitas Research Hospital, Rozzano, 20089 Milan, Italy; 2Department of Biomedical Sciences, Humanitas University, Pieve Emanuele, 20090 Milan, Italy; 3Fondazione Livio Sciutto Onlus, Campus Savona, Università degli Studi di Genova, 17100 Savona, Italy; 4Harvey Research Institute, Queen Mary University of London, Charterhouse Square, London EC1M 6BQ, UK

**Keywords:** periprosthetic joint infections, hip, knee, pentraxin 3, biomarkers

## Abstract

Background: Preoperative diagnosis of periprosthetic joint infections (PJIs) poses an unmet clinical challenge. The long pentraxin PTX3 is a component of the innate immune system involved in infection immunity. This study evaluated the potential of synovial and plasmatic PTX3 in the diagnosis of hip and knee PJIs. Methods: Consecutive total hip and knee arthroplasty (THA/TKA) revisions were prospectively included and classified as septic or aseptic according to the European Bone and Joint Infection Society (EBJIS) and Musculoskeletal Infection Society (MSIS) criteria. The concentration of PTX3 in plasma and synovial fluid samples was measured with ELISA. The AUC, threshold value, sensitivity, specificity, and positive and negative likelihood ratios were calculated using the ROC (receiver operating characteristic) curve method. Results: The study population included 128 patients (94 THAs; 34 TKAs). The AUC of the synovial PTX3 based on EBJIS criteria was 0.85 (*p* < 0.0001), with a sensitivity of 81.13% and a specificity of 93.33%. The AUC based on MSIS criteria was 0.95 (*p* < 0.001), with a sensitivity of 91.43% and a specificity of 89.25%. Plasmatic PTX3 failed to discriminate infected from non-infected patients. Conclusions: Synovial PTX3 demonstrated an excellent diagnostic potential in hip and knee PJIs, with a very high specificity irrespective of the diagnostic criteria for PJI.

## 1. Introduction

Periprosthetic joint infections (PJIs) are the most common cause of total hip and knee arthroplasty (THA and TKA, respectively) revision within two years of surgery, accounting for 62% of revisions in the biennium 2017–2018 [1,2]. Moreover, PJIs are the second cause of THA and TKA revision, after aseptic loosening, regardless of time since surgery. Given that the reoperation rate is 1.6% to 2.4% [1,2], the incidence of PJIs lies in the 1–1.5% range. According to the Swedish Hip and Knee Replacement Registries, 30,404 TKAs and 36,791 THAs have been performed in the biennium 2017–2018 [1,2]. Therefore, a significant number of patients develop septic complications after replacement surgery every year. Preoperative diagnosis of PJI is still challenging. Over time, several classification systems have been developed to formulate a proper diagnosis of PJI, such as the Musculoskeletal Infection Society (MSIS) [3,4], International Consensus Meeting (ICM) [5], Infectious Diseases Society of America (IDSA) [6], and European Bone and Joint Infection Society (EBJIS) [7,8,9,10,11] criteria. Confirmation of diagnosis requires isolation of the pathogen from explanted prosthesis and/or biopsies and its identification through microbiological techniques. On the other hand, preoperative diagnosis is highly desirable to inform and orient the clinical management of PJI patients prior to revision surgery. Preoperative diagnostic criteria integrate varying levels of information that are contributed by a number of clinical and biochemical parameters and have been consistently upgraded over recent years.

In this regard, several biomarkers have been assessed for diagnosis of PJIs, in particular those present in the synovial fluid and serum [12]. A major research focus has been on the relationship between infection and soluble mediators of the innate immune system [13,14]. At a systemic level, C-reactive protein (CRP), a paradigmatic component of the acute phase reaction, has been extensively studied as a serum biomarker of PJI, in conjunction with characteristics of haemostasis, such as erythrocyte sedimentation rate (ESR) and plasmatic concentration of D-dimer [5]. In addition to circulating biomarkers of PJI, molecular and cellular components of the synovial fluid, including α-defensin and leukocyte esterase (LE), which act as local “signs” of pathology, have received great attention [14]. Moreover, evidence is being accumulated that additional players of the innate immune system are involved in the pathogenesis of PJIs [12], which makes them suitable candidates for the development of novel diagnostic tools.

The long pentraxin 3 (PTX3) is a member of the pentraxin superfamily of proteins that includes CRP and Serum Amyloid P component (SAP, an acute phase protein in rodents), and is characterized by the presence of an N-terminal domain in addition to a C-terminal pentraxin-like domain (distinctive of the family) [15]. Folded into a glycosylated homo-octamer with a unique quaternary structure [16,17], this protein is made and released by different cell types such as mononuclear phagocytes, dendritic cells, smooth muscle cells, adipocytes, and fibroblasts [18,19,20,21,22]. The expression of PTX3 is induced by pro-inflammatory signals, such as IL-1β, TNF-α, and Toll-like receptors (TLR) agonists [15,23,24], via both MyD88- and TRIF-dependent pathways [25]. Gene transcription and protein synthesis are also elicited through PI3K/Akt and JNK pathways, or downstream of FUS/CHOP translocation [15]. Moreover, the protein is stored in the lactoferrin+ granules of neutrophils and is rapidly mobilized and secreted upon stimulation with micro-organisms, TLR agonists, and pro-inflammatory cytokines [26]. Neutrophils are promptly recruited to sites of infection and inflammation, and the rapid release of neutrophil-stored PTX3 is taken as an early immediate mechanism of innate defence in that this protein acts as an “opsonin” that facilitates pathogen recognition, internalization, and killing by professional phagocytes (including neutrophils themselves) [27,28]. Importantly, this long pentraxin is an intrinsic component of the bone microenvironment, where it is expressed by osteoprogenitor cells and is involved both in bone homeostasis and fracture healing [29,30].

Similarly to the close relative CRP, the plasmatic levels of PTX3 rapidly increase in several pathological conditions with inflammatory and/or infectious aetiology, including acute myocardial infarction, sepsis, and SARS-CoV-2 infections, and have been associated with severity of the disease and risk of mortality [31,32,33,34,35]. In most cases, PTX3 correlates with CRP; however, while CRP is mostly synthesized by hepatocytes and is extensively used as a systemic, though generic, marker of inflammation, PTX3 is locally induced at sites of infection and inflammation by proinflammatory mediators and is predicted to be an earlier and local marker of disease [36]. The circulating levels of PTX3 also correlate with other markers of bacterial infections, such as procalcitonin (PCT) and D-dimer, as recently reported in septic patients [37,38]. However, both PCT and D-dimer had limited accuracy as PJI markers [39], and their diagnostic potential in this pathology requires further investigation. Therefore, PTX3, at the interface between infection immunity and bone biology, stands out as an ideal candidate for investigations into new biomarkers of PJI.

Based on this rationale, the primary aim of this study was to assess the diagnostic potential of synovial and plasmatic PTX3 in PJI patients undergoing THA or TKA revision. In this regard, previous studies have shown that the diagnostic accuracy of an established PJI biomarker, i.e., alpha-defensin, depends on the clinical definition of infection [40]. To account for this, in our investigations, two internationally recognized and clinically validated criteria were adopted to define the infection status of the study population, based on the indications of MSIS [4] and EBJIS [7,8].

## 2. Materials and Methods

### 2.1. Study Design and Sample Size

This study complied with the provisions of the Declaration of Helsinki and was approved by the Institutional Review Board of IRCCS Humanitas Research Hospital (registration number: 165/2017). Confidentiality of patient data was preserved, and no patient identifiers were used in the dataset. Patients were enrolled only after obtaining a signature of written informed consent. The present prospective diagnostic study was performed at our institution from October 2016 to December 2019, following the International Standards for Reporting Diagnostic Accuracy (STARD) guidelines. The minimum follow-up was 24 months.

The primary aim of this study was to assess the performance of PTX3 (i.e., the concentration of the protein in the synovial fluid and plasma) in the diagnosis of PJI in patients undergoing total THA or TKA revision. As secondary aims, the diagnostic power of PTX3 (either in the synovial fluid or plasma) was compared with that of plasmatic CRP, and correlations were assessed between the concentration of PTX3 and that of other clinically established inflammatory markers.

A total of 128 patients were enrolled in this study, of which 53/35 were infected and 75/93 were non-infected (based on EBJIS/MSIS criteria). In this regard, it is worth noting here that in a pilot investigation on 40 patients receiving THA or TKA revision at IRCCS Humanitas Research Hospital, the concentration of PTX3 in the synovial fluid was able to detect PJI with an AUC of 0.93 (95% confidence interval—CI, 0.84–1.0). Given the primary aim of the study and assuming an AUC of 0.95 and a marginal error of 0.05 with 95% CI, the required number of patients to achieve a statistical power of 80% was estimated as 50 in both infected and non-infected groups.

### 2.2. Study Population

Patients eligible for THA and TKA revision surgery were enrolled prospectively and consecutively. To be included in the study, patients had to meet the following inclusion criteria: painful THA or TKA for at least 3 months, and sufficient clinical and laboratory data to define the presence or absence of PJI. Exclusion criteria included: use of antibiotics, glucocorticoids, and anti-histaminic therapy in the 2 weeks prior to surgery; rheumatoid arthritis and other rheumatic disorders; revision surgery for spacer removal and reimplantation; metallosis, prosthetic dislocation, periprosthetic fracture, limb length discrepancy, prosthetic rupture, and polyethylene wear.

Revisions were classified as septic or aseptic according to the MSIS [4] and EBJIS criteria [7,8] (Table 1).

For the MSIS criteria, elevated CRP was defined as >10 mg/L or >100 mg/L in chronic or acute infections, respectively; elevated synovial fluid leukocyte count was defined as >3000 leukocytes/mL or >10,000 leukocytes/mL in chronic or acute infections, respectively; elevated synovial fluid percentage of granulocytes was defined as >80% or >90% in chronic or acute infections, respectively; and positive histological analysis of periprosthetic tissue was defined as >5 neutrophils per high-power field (HPF) in 5 HPFs observed on periprosthetic tissue at ×400 magnification. For the EBJIS criteria, increased synovial fluid leukocyte count was defined as a leukocyte count of >2000/mL or >70% granulocytes; positive histopathology was defined as a mean of >23 granulocytes per 10 HPFs [41]; confirmatory microbial growth in synovial fluid and periprosthetic tissue culture was considered positive if ≥1 specimen was positive for highly virulent organisms (e.g., *Staphylococcus aureus*) or ≥2 specimens were positive for low virulent pathogens, and sonication culture was considered positive if >50 colony-forming units (CFU)/mL of sonicated fluid were counted [42].

### 2.3. Description of Treatment

Preoperative assessment of patients was based on physical examination, laboratory tests, including CRP and ESR, and plain radiographs including anterior–posterior (AP) views of the pelvis and axial views of the hip for THA, AP and lateral views of the knee, axial views of patella, and full-length weight-bearing views of bilateral lower extremities for TKA. Bone scintigraphy, CT scan, or MRI were performed according to the surgeon’s indications. Preoperative joint aspiration of synovial fluid for leukocyte counting and microbiology culturing was performed in case of high levels of CRP and/or ESR, or in the presence of high clinical suspicion for PJI based on multiple surgery, history of surgical site infection in the index joint or prior PJI.

All procedures were performed using the posterolateral approach with the patient in lateral decubitus for the hip surgery, and through the standard medial parapatellar approach for the knee surgery. The antibiotic prophylaxis with cefazolin or clindamycin was administered in all patients before surgery. Blood samples were withdrawn into EDTA Vacutainer immediately before surgery, from which plasma was isolated (see the Section 2.4 below) and used to measure the plasmatic concentration of the PTX3 protein. Synovial fluid was collected during surgery before capsulotomy to prevent blood contamination. Synovial fluid samples were used for microbiological and laboratory analyses. During surgery, five to seven intra-operative periprosthetic tissue samples were obtained for microbiological and histological analysis. Presence and identity of aerobic and anaerobic bacteria and fungi was assessed through microbiological cultures (14 to 21 days) of tissue extracts in selective media. The removed prosthesis was sonicated.

The enrolled patients underwent either one-stage or two-stage revision according to presence/absence of a suspicion of PJI (based on preoperative workout). The measured concentrations of synovial and plasmatic PTX3 were not provided to the surgeon at the time of surgery; therefore, they did not affect diagnosis of PJI and the clinical post-surgery management of patients.

After surgery, all patients received an empiric antibiotic treatment with vancomycin and piperacillin-tazobactam, vancomycin and ciprofloxacin or levofloxacin, or cefazolin until microbiological results were available. The antibiotic regimen was chosen on the basis of known patient risk factors and intolerance to penicillin. The empiric antibiotic treatment was interrupted after 7 days in patients without evidence of PJI. In patients with PJI, the antibiotic treatment was aetiologic in culture-positive cases and empiric in culture-negative cases. The antibiotic treatment was performed until reimplantation in patients who underwent two-stage revision. In patients having one-stage revision, the duration of the antibiotic therapy was three months.

### 2.4. Laboratory Analysis

All plasma and synovial fluid samples were sent to the laboratory on the day of surgery. Venous blood samples were collected in the presence of EDTA and centrifuged within 2 h at 2000 rpm for 10 min at room temperature. Plasma and synovial fluids were stored at −80 °C until use. Synovial and plasmatic PTX3 were measured by Sandwich ELISA using a home-made colorimetric assay, as previously described [43]. In this assay, a signal was obtained from the chromogenic substrate 3–3′ 5–5′ tetramethylbenzidine (TMB, 1-step ultra TMB-ELISA; Thermo Scientific, Waltham, MA, US), and colour development was stopped by addition of sulfuric acid. Absorbance was measured at 450 nm using an automated microplate reader (Versamax, Molecular Device Corporation, San Jose, CA, US). The concentration of PTX3 was calculated based on a standard curve of the recombinant protein (linear range: 75 pg/mL to 2.4 ng/mL) using the SoftMax PRO software (version 5.3; Molecular Device Corporation, San Jose, CA, US). Plasmatic CRP levels were quantified by immunoturbidimetry on a Beckman Coulter instrument (Beckman Coulter, Milan, Italy), while D-dimer was measured according to routine protocols using the ACL-TOP 750 LAS (Werfen, Milan, Italy) coagulation analyser. ESR and synovial fluid leukocyte count were measured by standard procedures in use in the Institutional Clinical Laboratory.

### 2.5. Statistical Analysis

Descriptive statistics and statistical testing were performed using Prism version 9.4.1 (GraphPad, Boston, MA, USA). In an initial round of analysis, normality of values measured for the synovial and plasmatic markers in the study (including PTX3; continuous variables) was assessed by means of the Shapiro–Wilk and Kolomogorov–Smirnov tests. The distribution of these observations across groups of patients (i.e., infected vs. non-infected) was evaluated with the Mann–Whitney test. In all cases, values were represented as median ± interquartile range (IQR). Categorical variables (e.g., sex, Charlson and ASA scores) were expressed as frequencies and analysed using contingency tables by employing the Pearson chi-square test to assess differences between groups. Correlations between continuous variables (i.e., concentration of PTX3 and leukocyte count in the synovial fluid) were assessed using the Spearman’s rank correlation coefficient method.

In a second round of analysis, a logistic regression model was fitted to the data using the concentration values of synovial and plasmatic markers as explanatory variables and the infection status (defined based on either MSIS or EBJIS criteria) as the response variable. ROC (receiver operating characteristic) curves were generated to assess the overall quality of fitting. For each synovial and plasmatic marker, threshold value, sensitivity, specificity, and positive and negative likelihood ratios were calculated using the corresponding ROC curves according to the Youden index (J). The area under the ROC curve (AUC) was then calculated and reported as a synthetic index of diagnostic performance of the selected markers.

## 3. Results

### 3.1. Study Population

From October 2016 to December 2019, a total of 544 patients who were referred to our institute for THA and TKA revision were classified as potentially eligible. Of these, 190 patients could not enter the study because they did not fulfil the MSIS and EBJIS criteria, and 161 were excluded according to the exclusion criteria. Of the remaining 193 eligible patients, 65 were excluded because of lack of synovial fluid to perform the index tests (Figure 1).

The study population consisted of 128 patients, including both THA (*n* = 94) and TKA (*n* = 34) revisions. A total of 53 patients had PJI according to the EBJIS criteria. Of these, 35 were deemed infected according to the MSIS criteria too. No patient with a diagnosis of PJI according to the MSIS criteria was deemed non-infected according to the EBJS criteria. All patients were comparable in age, gender, BMI, Charlson score, and ASA scores (Table 2).

### 3.2. Microbiological Analysis

The most frequent pathogens isolated in the infected patients (Table 3) were the coagulase-negative Staphylococci (15 and 13 patients in the EBJIS- and MSIS-positive groups, respectively), followed by Staphylococcus Aureus (7 in both groups) and Streptococci (5 and 4). The percentage of poly-microbial infections was 11% and 9% in the EBJIS- and MSIS-positive groups, respectively.

### 3.3. Laboratory Analysis

Synovial PTX3 and plasmatic CRP levels were significantly higher in the infected patients compared to the non-infected ones when either EBJIS or MSIS criteria were considered for diagnosis of PJI (*p* < 0.0001). No differences were found for plasmatic PTX3, ESR, and D-dimer (Table 4, Figure 2, and Appendix A).

When patients were stratified based on the explanted implant (THA or TKA), the synovial levels of PTX3 remained significantly higher in the infected individuals compared to the ones who had aseptic revision, regardless of the criteria used for diagnosis of PJI (Figure 3).

To better assess the diagnostic power of PTX3, a logistic regression analysis was run on the datasets generated in this study, which allowed calculation of the AUC and other testing parameters. In this regard, the AUC of synovial PTX3 was 0.85 (95%IC, 0.78–0.93, *p* < 0.0001), with a sensitivity of 81.13% and a specificity 93.33% when using EBJIS criteria for diagnosis of PJI. When the MSIS criteria were considered, the diagnostic power of synovial PTX3 was even higher, with an AUC of 0.95 (95%IC, 0.91–0.98, *p* < 0.001), a sensitivity of 91.43%, and a specificity of 89.25% (Table 5). As a comparison, the AUC of plasmatic CRP, a clinically established biomarker of PJI, was 0.80 (95%IC, 0.72–0.88, *p* < 0.0001), with a sensitivity of 62.26% and a specificity of 80% (according with EBJIS), or 0.81 (95%IC, 0.72–0.89, *p* < 0.0001), with a sensitivity of 71.43% and a specificity of 75.27% (according with MSIS) (Table 5 and Appendix A). Furthermore, synovial PTX3 retained elevated diagnostic performance regardless of sites of revision surgery (either TKA or THA; Appendix A, and Appendix A).

Plasmatic PTX3, ESR, and D-dimer proved poor diagnostic markers and failed to discriminate infected from non-infected patients (regardless of the criteria used for diagnosis of PJI) (Table 5).

ROC curves of synovial and plasmatic PTX3 are shown in Figure 4 and Appendix A. For synovial PTX3, the best concentration values to discriminate between infected and aseptic patients was 4.1 ng/mL and 5.7 ng/mL for EBJIS and MSIS criteria, respectively.

The leukocyte count in the synovial fluid was significantly higher in the infected revisions compared with aseptic revisions (*p* < 0.0001), according with the MSIS criteria. Moreover, the concentration of PTX3 was significantly correlated with leukocyte count in the synovial fluid.

### 3.4. Clinical Outcome

Among the 53 patients with a diagnosis of PJI according to the EBJIS criteria, 29 underwent two-stage revision, 12 total one-stage revision, and 12 partial one-stage revision. The treatment failed in five patients: three individuals (two two-stage and one one-stage revision) underwent long-term antibiotic suppression treatment, one patient underwent surgical debridement (DAIR) for peri-operative infection after the two-stage revision, and one patient underwent further revision for septic acetabular loosening after the one-stage partial revision. Among the 75 aseptic patients, 3 underwent two-stage revision, 46 total one-stage revision, and 26 partial one-stage revision. No patients reported the failure of treatment.

## 4. Discussion

Periprosthetic joint infections (PJIs) represent one of the most common causes of joint replacement failure [1,2]. Although several classification systems have been developed to formulate a proper diagnosis, the diagnosis of PJIs is still challenging, particularly in septic cases associated with low-virulence and biofilm-forming pathogens. Accurate preoperative diagnosis is needed to choose proper antibiotic therapies and surgical strategies. Indeed, undetected and/or mistreated PJIs at the time of revision surgery can result in persistence of the infection, failure of the revised implant, longer hospital stays, multiple surgeries, prolonged immobilization and rehabilitation, together with increased overall costs. To support and orient diagnosis of PJI prior to surgery it is therefore timely to identify and validate novel biomarkers.

The long pentraxin PTX3 is an established humoral component of the innate immune system and an emerging player in bone biology [29,30]. This protein is synthesized and released at sites of infection/inflammation by a number of immune and non-immune cells, including macrophages, dendritic cells, neutrophils, and cells of the osteoblastogenic lineage, after stimulation with primary proinflammatory cytokines (e.g., TNF-α and IL-1β) and microbial components (e.g., LPS) [22,23,24,26]. The locally made protein binds a broad spectrum of microorganisms, including selected fungi, viruses, and bacteria, and exerts a number of host-protective functions [27,35]. Previous studies demonstrated that high levels of plasmatic PTX3 are associated with risk and severity of several inflammatory and infective diseases [31,32,33,34,35]. Therefore, the present study was designed to assess the potential of synovial and plasmatic PTX3 as diagnostic biomarker of PJIs in patients undergoing THA or TKA revision. The main finding of our investigation was that the concentration of PTX3 in the synovial fluid was elevated in THA/TKA patients with PJI (compared to those who had aseptic prosthesis revision) and able to predict the infection with high accuracy (i.e., AUC values of 0.85 and 0.95 for EBJIS and MSIS criteria, respectively). Importantly, the specificity of synovial PTX3 was consistently high across the applied diagnostic classifications (93% and 89% for EBJIS and MSIS, respectively), which suggests that measuring the levels of this long pentraxin in the synovial fluid might provide clinically useful information to confirm diagnosis of PJI rather than support screening investigations. As opposed to the synovial protein, the plasmatic concentration of PTX3 had poor diagnostic value, likely due to this long pentraxin being mainly synthesized in loco at sites of infection, unlike the short pentraxin CRP that is systemically produced by the liver in response to IL-6. In a previous study, Mauri et al. [44] demonstrated that a PTX3 level ≥ 1 ng/mL in the bronchoalveolar lavage fluid was discriminative of microbiologically confirmed pneumonia in mechanically ventilated patients. On the other hand, the plasmatic PTX3 was not effective for the diagnosis. Consistent with this view, a correlation was found in our study between leukocyte count and concentration of PTX3 in the synovial fluid, which suggests that the latter is likely contributed by white blood cells (mostly neutrophils) infiltrating the inflamed synovium during the infection, where these cells are known to be a source of PTX3 in inflammatory conditions (Figure 5).

In a previous meta-analysis, Wyatt et al. [14] reported a pooled diagnostic sensitivity and specificity of alpha-defensin for PJI of 1.00 (95% CI, 0.82 to 1.00) and 0.96 (95% CI, 0.89 to 0.99), respectively. They also observed a pooled diagnostic sensitivity and specificity of leukocyte esterase for PJI of 0.81 (95% CI, 0.49 to 0.95) and 0.97 (95% CI, 0.82 to 0.99), respectively. In a more recent meta-analysis, Kuiper et al. [45] compared the diagnostic potential of laboratory-based (ELISA) and lateral-flow (LF) alpha-defensin in a pooled cohort that included THA and TKA patients. The authors did not observe significant differences between ELISA and LF alpha-defensin in terms of sensitivity (90% versus 86%) and specificity (97% versus 96%). In both studies, the original or modified MSIS criteria were used as the reference standard for the diagnosis of PJI. The diagnostic potential of synovial PTX3, as assessed in the present study, is therefore in line with that of other synovial inflammatory markers; however, a thorough comparison would require further studies in homogenous cohorts of patients.

No serum markers have been shown to be accurate enough to diagnose PJIs in THA/TKA patients (i.e., AUC values of serum markers have been consistently documented to be below 70% [39]). In this regard, the positive likelihood ratio of serum markers has been reported to be below six points, indicating a <35% increase in probability of finding an infection [40]. Therefore, only the synovial fluid could provide useful and reliable information for diagnosis of PJIs [12]. Compared to the previous literature [9,12,13,14,45], the present study further corroborates the notion that serum markers have a limited role in the detection of PJIs and cannot be used on their own to assess septic patients. On the other hand, to support diagnosis of PJI, the synovial fluid needs to be gathered when the infection is suspected.

Due to the large variety of pathogens found in the specimens analysed in this study, we could not make any statistically sound association between the levels of PTX3 and the identity of the pathogens. However, it is licit to speculate that highly virulent pathogens can strongly stimulate the immune system and therefore induce high levels of PTX3. In this respect, both in preclinical models and in patients, PTX3 plasma levels rapidly increase in response to different infectious agents, including bacteria, viruses, and fungi, and correlate with disease severity and unfavourable outcomes [35].

A limitation of the present study is that the reference standards used for diagnosis of PJI (MSIS and EBJIS criteria) did not include synovial alpha-defensin. In fact, our study was authorized and commenced prior to publication of the latest (alpha-defensin including) version of the EBJIS and ICM criteria. Consequently, we could not make any direct comparison between synovial alpha-defensin and PTX3 as for their diagnostic potential in PJI. Another limitation is that patients with clear or suspected auto-inflammatory conditions were excluded based on the reasoning that sterile activation of the immune system could have potentially resulted in elevation of the PTX3 levels in the synovial fluid and/or blood regardless of the presence of PJI. On the same line, patients undergoing prosthetic revision due to metallosis or severe wear of the polyethylene were not eligible. Further studies should be performed to investigate the diagnostic potential of PTX3 in these subgroups of patients.

## 5. Conclusions

Synovial PTX3 demonstrated an excellent diagnostic potential in hip and knee PJIs, with a very high specificity regardless of the criteria used for diagnosis of PJI. This points to PTX3 being a useful marker to confirm a suspected infection. Further studies are needed to develop and investigate the diagnostic performance for PJI of a new classification system that includes synovial PTX3.

## Figures and Tables

**Figure 1 jcm-12-01055-f001:**
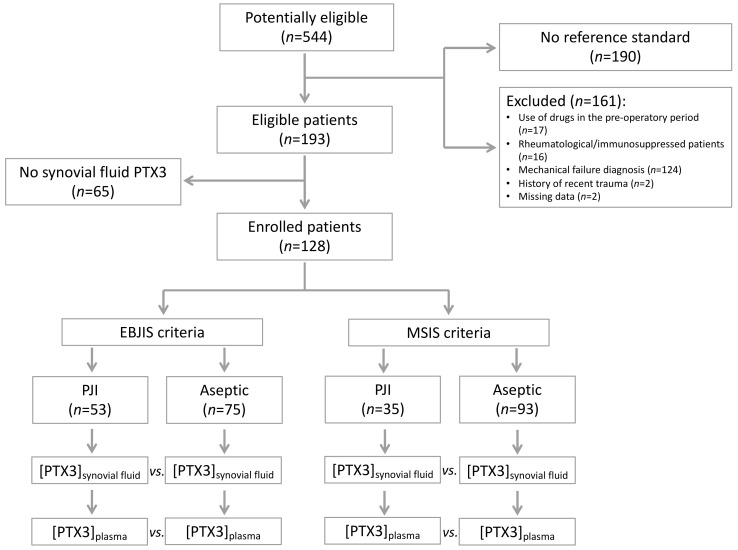
Flow-chart of the patients included in the study.

**Figure 2 jcm-12-01055-f002:**
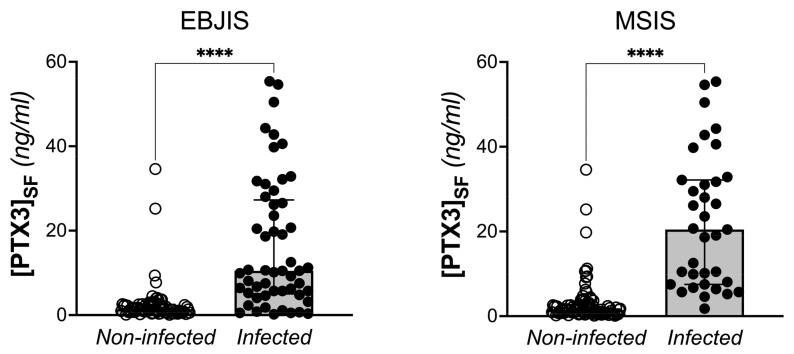
Concentration of PTX3 in the synovial fluid of THA and TKA patients with and without PJI, based on EBJIS and MSIS criteria (Mann-Withney test, **** *p* < 0.0001).

**Figure 3 jcm-12-01055-f003:**
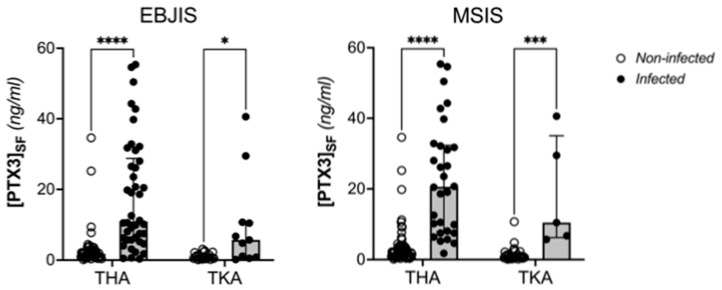
Distribution of the concentration of PTX3 in the synovial fluid of aseptic and infected patients according to surgery (TKA or THA) and EBJIS/MSIS criteria (Two Way ANOVA, **** *p* < 0.0001, *** *p* < 0.0005, * *p* < 0.05).

**Figure 4 jcm-12-01055-f004:**
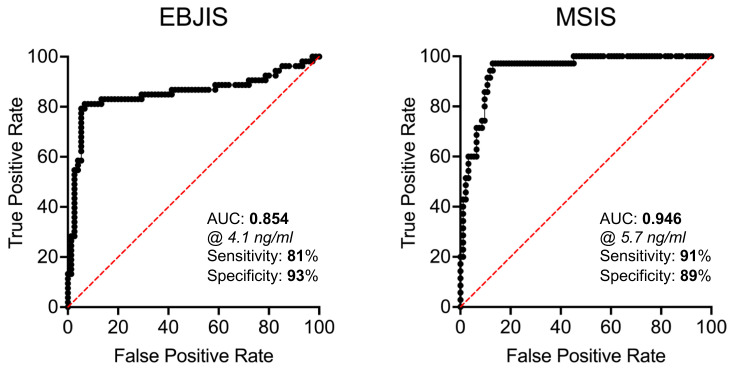
ROC curves of the concentration of PTX3 in the synovial fluid of THA and TKA patients with and without PJI (based on EBJIS and MSIS criteria).

**Figure 5 jcm-12-01055-f005:**
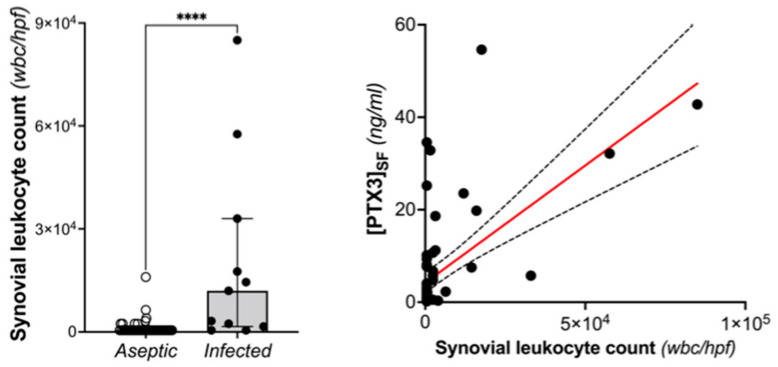
Leukocyte count in the synovial fluid of THA and TKA patients with and without PJI (based on MSIS criteria; Mann-Withney, **** *p* < 0.0001) and its correlation with the concentration of PTX3 (Spearman r: 0.498, *p* < 0.0001).

**Table 1 jcm-12-01055-t001:** Definition of periprosthetic joint infections.

MSIS (1/2 Major Criteria or 3/5 Minor Criteria)	EBJIS (1/4 Criteria)
Major criteria:	Purulence around the prosthesis or sinus tract
Two positive periprosthetic cultures	Increased synovial fluid leukocyte count
Sinus tract communicating with the prosthesis	Positive histopathology
Minor criteria:	Confirmatory microbial growth in synovial fluid, periprosthetic tissue, or sonication culture
Elevated CRP and ESR	
Elevated synovial fluid leukocyte count or positive leukocyte esterase strip test (++ or +++)	
Elevated synovial fluid percentage of granulocytes	
A single positive culture	
Positive histological analysis of periprosthetic tissue	

MSIS, Musculoskeletal Infection Society; EBJIS, European Bone and Joint Infection Society; CRP, C-reactive protein; ESR, erythrocyte sedimentation rate.

**Table 2 jcm-12-01055-t002:** Demographics of the study population.

		EBJIS			MSIS		
	All patients (*n* = 128)	Infected (*n* = 53)	Non-infected (*n* = 75)	*p*	Infected (*n* = 35)	Non-infected (*n* = 93)	*p*
Sex (M:F)	51:77	26:27	25:50	0.074 ^b^	17:18	34:59	0.216 ^b^
BMI	28.37 ^a^ (24.54–30.94)	29.14 (24.23–32.04)	27.13 (24.60–29.66)	0.163 ^c^	29.14 (24.08–33.38)	27.27 (24.65–29.76)	0.144 ^c^
Charlson score	0: 11 (8%) 1: 16 (13%) 2: 31 (24%) 3: 42 (33%) 4: 17 (13%) 5: 9 (7%) 6: 2 (2%)	0: 6 (11%) 1: 9 (17%) 2: 15 (28%) 3: 12 (23%) 4: 6 (11%) 5: 4 (8%) 6: 1 (2%)	0: 5 (7%) 1: 7 (9%) 2: 16 (21%) 3: 30 (40%) 4: 11 (15%) 5: 5 (7%) 6: 1 (1%)	0.416 ^b^	0: 5 (14%) 1: 8 (23%) 2: 5 (14%) 3: 10 (29%) 4: 4 (11%) 5: 3 (9%) 6: 0	0: 6 (6%) 1: 8 (9%) 2: 26 (28%) 3: 32 (35%) 4: 13 (14%) 5: 6 (6%) 6: 2 (2%)	0.162 ^b^
ASA score	1: 22 (17%) 2: 72 (56%) 3: 34 (27%)	1: 9 (17%) 2: 30 (57%) 3: 14 (26%)	1: 13 (17%) 2: 42 (56%) 3: 20 (27%)	0.998 ^b^	1: 6 (17%) 2: 19 (54%) 3: 10 (29%)	1: 16 (17%) 2: 53 (57%) 3: 24 (26%)	0.948 ^b^
Age at surgery	70 (61–76) ^a^	69 (56–73)	70 (62–78)	0.242 ^c^	70 (52–74)	70 (62–77)	0.165 ^c^

^a^ Median (IQR); ^b^ Chi-square; ^c^ Mann–Whitney. BMI, body mass index; ASA, American Society of Anesthesiologists.

**Table 3 jcm-12-01055-t003:** Pathogens isolated at the microbiological analysis after total hip or knee arthroplasty revision.

Pathogens	PJI (EBJIS) *n* (Hip:Knee)	PJI (MSIS) *n* (Hip:Knee)
Coagulase Negative Staphylococci	15 (14:1)	13 (12:1)
Staphylococcus Aureus	7 (6:1)	7 (6:1)
Streptococcus Species	5 (4:1)	4 (3:1)
Pseudomonas Aeruginosa	2 (2:0)	2 (2:0)
Enterococcus Faecalis	2 (1:1)	1 (1:0)
Enterococcus Faecium	2 (1:1)	1 (1:0)
Corynebacterium Sp.	1 (0:1)	-
Microbacterium Sp.	1 (1:0)	-
Propionibacterium Acnes	1 (1:0)	-
Ralstonia Pickettii	1 (1:0)	-
Aspergillus Fumigatus	1 (1:0)	-
Brevibacterium Casei	1 (0:1)	-
Brevibacterium Laterosporus	1 (0:1)	-
Enterobacter Cloacae	1 (0:1)	-
Escherichia Coli	1 (0:1)	-
Not identified	16 (12:4)	9 (7:2)

*n*, number.

**Table 4 jcm-12-01055-t004:** Concentration of synovial and plasmatic biomarkers.

		EBJIS			MSIS		
	All Patients (*n* = 128)	Infected (*n* = 53)	Non-Infected (*n* = 75)	*p* ^b^	Infected (*n* = 35)	Non-Infected (*n* = 93)	*p*
Synovial PTX3 (ng/mL)	2.41 (0.91–9.33) ^a^	10.48 (5.02–27.27)	1.40 (0.76–2.52)	<0.0001	20.46 (7.51–32.15)	1.51 (0.74–2.79)	<0.0001
Plasmatic PTX3 (ng/mL)	4.89 (3.59–6.43)	4.66 (3.28–6.34)	5.25 (3.76–6.46)	0.347	4.66 (3.36–6.21)	5.23 (3.61–6.46)	0.567
Plasmatic CRP (ng/mL)	0.65 (0.22–1.61)	1.33 (0.76–3.34)	0.27 (0.16–0.90)	<0.0001	2.06 (0.88–4.26)	0.40 (0.17–0.94)	<0.0001
ESR (mm/h)	20.00 (10.00–38.75)	20.00 (10.25–55.75)	20.00 (9.25–34.00)	0.200	22.00 (11.50–61.00)	20.00 (9.00–34.00)	0.105
D-Dimer (ng/mL)	325 (249–449)	318 (247–503)	342 (226–440)	0.936	347 (241–610)	325 (252–413)	0.433

^a^ Median (IQR); ^b^ Mann–Whitney. PJI, periprosthetic joint infection; MSIS, Musculoskeletal Infection Society; EBJIS, European Bone and Joint Infection Society; CRP, C-reactive protein; ESR, erythrocyte sedimentation rate.

**Table 5 jcm-12-01055-t005:** ROC curve analysis of synovial and plasmatic biomarkers.

EBJS	Obs ^a^	Threshold ^b^	Sensitivity	Specificity	Accuracy	SE ^c^	LR+ ^d^	LR- ^e^	AUC	*p*
Synovial PTX3 (ng/mL)	128	4.1	81.13%	93.33%	87.5%	0.04	11.89	0.22	0.85 (0.78–0.93) ^f^	<0.0001
Plasmatic PTX3 (ng/mL)	124	4.7	54.00%	59.46%	57.45%	0.05	1.28	0.72	0.55 (0.45–0.66)	0.345
Plasmatic CRP (mg/dL)	128	0.94	62.26%	80.00%	72.66%	0.04	3.23	0.49	0.80 (0.72–0.88)	<0.0001
ESR (mm/h)	96	40	36.11%	86.67%	67.71%	0.06	2.71	0.74	0.58 (0.45–0.70)	0.198
D-Dimer (ng/mL)	33	516	25.00%	82.24%	57.58%	0.10	2.13	0.85	0.51 (0.31–0.71)	0.928
**MSIS**	**Obs**	**Threshold**	**Sensitivity**	**Specificity**	**Accuracy**	**SE**	**LR+**	**LR-**	**AUC**	** *p* **
Synovial PTX3 (ng/mL)	128	5.7	91.43%	89.25%	89.84%	0.02	8.50	0.10	0.95 (0.91–0.98)	<0.001
Plasmatic PTX3 (ng/mL)	124	4.8	54.44%	55.88%	50.81%	0.06	0.77	1.17	0.53 (0.42–0.65)	0.564
Plasmatic CRP (mg/dL)	128	0.94	71.43%	75.27%	74.22%	0.04	2.90	0.41	0.81 (0.72–0.89)	<0.0001
ESR (mm/h)	96	40	42.86%	84.00%	75.00%	0.07	2.68	0.68	0.62 (0.47–0.76)	0.104
D-Dimer (ng/mL)	33	516	40.00%	91.30%	75.76%	0.11	4.60	0.66	0.59 (0.36–0.81)	0.422

^a^ Number of observations; ^b^ threshold unit; ^c^ standard error; ^d^ positive likelihood ratio; ^e^ negative likelihood ratio; ^f^ area (95% CI). MSIS; Musculoskeletal Infection Society; EBJIS; European Bone and Joint Infection Society; CRP, C-reactive protein; ESR, erythrocyte sedimentation rate.

## Data Availability

The data supporting reported results can be found in a repository (Zenodo).

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
