# Peer review of "Long Pentraxin 3 as a New Biomarker for Diagnosis of Hip and Knee Periprosthetic Joint Infections"

_jcm, 2023, doi:10.3390/jcm12031055_

Round 1

Reviewer 1 Report

Thank you for the opportunity to review this manuscript. The authors conducted a prospective diagnostic study to evaluate the performance of PTX3 in the diagnosis of PJI. Both THA or TKA revision were included. Based on their results, the authors concluded that PTX3 may improve the diagnostic accuracy for PJI. Overall, the work is interesting and stimulating which would add new evidence on PJI. However, some minor issues should be addressed.

1.      Please provide the trial registration number and ethics number.

2.      Please clarify the time of sample collection (immediately after admission?).

3.      Please clarify the detecting instrument (the brand and model).

4.      Did the authors collect the intraoperative tissue samples for standard culture and extended culture?

5.      The authors should evaluate and interpret all data separately (THA or TKA revision).

6.      Please include the types of bacteria found in each of the groups (THA or TKA revision).

Author Response

Dear Editor,

Please find attached the revised version of the above manuscript. The comments of the reviewers have been carefully considered, and implemented as follows. Please note that all changes made in the document (including those related to the points raised by Reviewers 1 and 2) have been highlighted in green to facilitate tracking and reading.

Reviewer 1

Thank you for the opportunity to review this manuscript. The authors conducted a prospective diagnostic study to evaluate the performance of PTX3 in the diagnosis of PJI. Both THA or TKA revision were included. Based on their results, the authors concluded that PTX3 may improve the diagnostic accuracy for PJI. Overall, the work is interesting and stimulating which would add new evidence on PJI. However, some minor issues should be addressed.

  1. Please provide the trial registration number and ethics number.

This information has been incorporated into the “Study Design and Sample Size” paragraph (lines 104-107) and Institutional Review Board Statement section (lines 438-440) of the revised manuscript.

  1. Please clarify the time of sample collection (immediately after admission?).

The blood samples were collected immediately before surgery (line 167), whereas the synovial fluid was collected during surgery before capsulotomy to prevent blood contamination (line 169).

  1. Please clarify the detecting instrument (the brand and model).

The “Laboratory Analysis” paragraph has been expanded to accommodate the missing technical details (lines 194-200), including brand and model of the detecting instrument, an explanation of how the concentration of PTX3 was worked out, and the software used for these calculations.

  1. Did the authors collect the intraoperative tissue samples for standard culture and extended culture?

Presence and identity of aerobic and anaerobic bacteria and fungi was assessed through microbiological cultures (14 to 21 days) of tissue extracts in selective media. This information has been incorporated in the revised manuscript (lines 172-174).

  1. The authors should evaluate and interpret all data separately (THA or TKA revision).

We have evaluated the THA and TKA data separately. The resulting ROC curves and diagnostic indexes are shown in Figures S3 and S4, and Table S1, respectively (Supplementary Information). Of note, synovial PTX3 retained elevated diagnostic performance regardless of sites of revision surgery (either TKA or THA), an observation that has been incorporated in the main text at lines 295-297.

  1. Please include the types of bacteria found in each of the groups (THA or TKA revision).

We included in the Table 3 the pathogens found in each group (THA and TKA revision). In the revision process we also changed some data entry errors.

Reviewer 2

The authors present a well-written manuscript about an unresolved problem in orthopaedic surgery. The search for reliable biomarkers to diagnose PJI is going on. The presented biomarker pentraxin 3 shows excellent statistical values to diagnose PJI. However, some questions have been raised while reading the manuscript.

  1. Up to now, synovial alpha-1-defensin AD-1 is the most frequently used biomarkers for diagnosis of PJI. Due to its high accuracy, it is part of the EBJIS criteria for the diagnosis of PJI. In this manuscript the EBJIS criteria are used to distinguish between PJI and aseptic complications. Why don’t you compare the values of PTX3 to AD-1 values?

A limitation of the present study is that the reference standards used for diagnosis of PJI (MSIS and EBJIS criteria) did not include synovial alpha-defensin. In fact, our study was authorized and commenced prior to publication of the latest (alpha-defensin including) version of the EBJIS and ICM criteria. Consequently, we could not make any direct comparison between synovial alpha-defensin and PTX3 as for their diagnostic potential in PJI. We have included this comment in the revised manuscript (lines 402-407).

  1. Line 42: Strong statement, it fits with my experience, do you have evidence for this?

We revised the statement as follows: “Confirmation of diagnosis requires isolation of the pathogen from explanted prosthesis and/or biopsies and its identification through microbiological techniques.” (lines 42-44)

  1. Line 62 ff. can you provide a pathway for the induction of Pentraxin 3?

The expression of PTX3 is induced downstream of signal transduction pathways that are involved in regulation of the inflammatory response, including engagement of TLRs and activation of PI3K/Akt or JNK. The most relevant ones are indicated in the revised version of the manuscript (lines 67-71 of the “Introduction” section). Also, we have modified the referencing to better clarify this point.

  1. Line 75: Pentraxin 3 is also elevated in myocardial infarction. So, it seems that PTX 3 is not only in expressed in infectious diseases. There are other biomarkers such as Procalcitonin which is elevated in inflammatory and infectious diseases. Please compare PTX 3 to such biomarkers.

The circulating levels of PTX3 are generally associated with the inflammatory response that accompanies infections and/or sterile tissue damage. As a matter of fact, the concentration of this pentraxin increases in different pathological conditions (some of which are listed in lines 82-83), including cardiovascular diseases, chronic kidney disease, bacterial, fungal and viral infections. Also, in these clinical settings PTX3 levels correlate with disease’s severity and are strong predictors of mortality. In many inflammatory/infectious conditions, a correlation has been documented between the plasmatic concentration of PTX3 and that of CRP, however, as pointed out in the manuscript (lines 80-81), the levels of PTX3 increase more rapidly than those of CRP. This is mainly due to the local production of PTX3 at sites of infection and tissue damage in response to primary proinflammatory cytokines and/or microbial components, as opposed to CRP whose synthesis mainly occurs in the liver in response to a rise in the circulating levels of IL-6. Regarding procalcitonin (PCT) and D-dimer, unfortunately we did not measure the synovial levels of these molecules in our cohort because they are not included in MSIS and EBJIS criteria. It is worth pointing out here that we and others have recently observed a correlation between the plasmatic levels of PTX3 and those of PCT and D-dimer in septic patients (Huan Chen et al., doi.org/10.3390/diagnostics11101906; Davoudian et al., doi.org/10.3389/fimmu.2022.979232). This information notwithstanding, available data indicate that these two molecules have limited potential in PJI diagnosis, essentially due to poor accuracy (Sigmund IK et al., doi.org/ 10.3390/biomedicines9091128). These points have been integrated in the “Introduction” section, which has been expanded accordingly (lines 84-92).

  1. Line 181: Which method was taken to measure CRP, D-dimer, and synovial fluid leukocyte count?

More indications on the methods used to quantify CRP and D-dimer are now included in section 2.4 Laboratory Analysis (lines 201-203).

  1. Line 352: The authors compare their results to reports about AD-1. There is only one reports about AD-1 presenting low sensitivity and high specificity to diagnose PJI. There are a lot more studies investigating AD-1 in the diagnostics of PJI. (for example https://www.ncbi.nlm.nih.gov/pmc/articles/PMC7319381/). Here you see much higher sensitivity values for AD-1. The reference 43 in the manuscript cannot be found in the reference list. Please provide reference 43 and choose also other reference to compare PTX 3 with AD-1.

We appreciate the Reviewer’s comment, and amedend the manuscript accordingly. In particular, we updated the referencing and presented more recent evidence as follows (lines 372-384): “In a previous meta-analysis, Wyatt et al. [45] reported pooled sensitivity and specificity for alpha-defensin in diagnosis of PJI of 1.00 (95% CI, 0.82 to 1.00) and 0.96 (95% CI, 0.89 to 0.99), respectively. They also observed pooled sensitivity and specificity for leukocyte esterase of 0.81 (95% CI, 0.49 to 0.95) and 0.97 (95% CI, 0.82 to 0.99), respectively. In a more recent meta-analysis, Kuiper et al. [46] compared the diagnostic potential of laboratory-based (ELISA) and lateral-flow-dosed (LF) alpha-defensin in a pooled cohort that included both THA and TKA patients. The authors did not observe significant differences between ELISA and LF alpha-defensin in terms of sensitivity (90% vs 86%) and specificity (97% vs 96%). In both studies, original and modified MSIS criteria were used as reference standard for diagnosis of PJI. The diagnostic potential of synovial PTX3, as assessed in the present study, is therefore in line with that of other synovial inflammatory markers, however a thorough comparison would require further studies in homogenous cohorts of patients.”

  1. Line 372: The fact that there is no gold standard for diagnosis of PJI is not a limitation of your study. It is the reason why you are performing this study. Please change this sentence.

We value the Reviewer’s point and recognize that this statement is rather misleading. We therefore omitted from the revised version of the manuscript.

We thank the Editorial Board and Reviewers for revising our manuscript. We appreciate your and the reviewer’s comments. We hope that the overall quality of the manuscript has improved, and is now amenable for publication in the Journal of Clinical Medicine.

Yours sincerely

Reviewer 2 Report

Review: JCM

Long Pentraxin 3 As A New Biomarker For Diagnosis Of Hip And Knee Periprosthetic Joint Infections

The authors present a well-written manuscript about an unresolved problem in orthopaedic surgery. The search for reliable biomarkers to diagnose PJI is going on. The presented biomarker pentraxin 3 shows excellent statistical values to diagnose PJI. However, some questions have been raised while reading the manuscript.

1. . Up to now, synovial alpha-1-defensin AD-1 is the most frequently used biomarkers for diagnosis of PJI. Due to ist high accuracy it is part of the EBJIS criteria for the diagnosis of PJI. In this manuscript the EBJIS criteria are used to distinguish between PJI and aseptic complications. Why don´t you compare the values of PTX3 to AD-1 values?

2. Line 42: Strong statement, it fits with my experience, do you have evidence for this?

3. Line 62 ff. can you provide a pathway for the induction of Pentraxin 3?

4. Line 75: Pentraxin 3 is also elevated in myocardial infarction. So, it seems that PTX 3 is not only in expressed in infectious diseases. There are other biomarkers such as Procalcitonin which is elevated in inflammatory and infectious diseases. Please compare PTX 3 to such biomarkers.

5. Line 181: Which method was taken to measure CRP  D-dimer, and synovial fluid leukocyte count ?

6. Line 352: The authors compare their results to reports about AD-1. There is only one reports about AD-1 presentin low sensitivity and high specifity to diagnose PJI. There are a lot more studies investigating AD-1 in the diagnostics of PJI. (for example https://www.ncbi.nlm.nih.gov/pmc/articles/PMC7319381/). Here you see much higher sensitivity values for AD-1. The reference 43 in the manuscript cannot be found in the reference list. Please provide reference 43 and choose also other reference to compare PTX 3 with AD-1.

7. Line 372: The fact that there is no gold standard for diagnosis of PJI is not a limitation of your study. It is the reason why you are performing this study. Please change this sentence.

Author Response

(The authors gave the same response as above.)

Round 2

Reviewer 2 Report

Thank you very much for the revised manuscript.

I only have one more question to point 1.

  1. Up to now, synovial alpha-1-defensin AD-1 is the most frequently used biomarkers for diagnosis of PJI. Due to its high accuracy, it is part of the EBJIS criteria for the diagnosis of PJI. In this manuscript the EBJIS criteria are used to distinguish between PJI and aseptic complications. Why don’t you compare the values of PTX3 to AD-1 values?

A limitation of the present study is that the reference standards used for diagnosis of PJI (MSIS and EBJIS criteria) did not include synovial alpha-defensin. In fact, our study was authorized and commenced prior to publication of the latest (alpha-defensin including) version of the EBJIS and ICM criteria. Consequently, we could not make any direct comparison between synovial alpha-defensin and PTX3 as for their diagnostic potential in PJI. We have included this comment in the revised manuscript (lines 402-407).

Could you please describe which EBJIS criteria version you took? The reader assumes you took the current ciriteria

Author Response

Manuscript ID: jcm-2129820

Type of manuscript:

Article Title: Long Pentraxin 3 As A New Biomarker For Diagnosis Of Hip And Knee Periprosthetic Joint Infections

Dear Editor,

Please find attached the revised version of the above manuscript. The comments of the reviewers have been carefully considered, and implemented as follows. Please note that all changes made in the document (including those related to the points raised by Reviewer 2) have been highlighted in green to facilitate tracking and reading.

Reviewer 2

  1. Could you please describe which EBJIS criteria version you took? The reader assumes you took the current ciriteria.

We thank the reviewer for giving us the opportunity to clarify this aspect. In the Table 1 were reported the MSIS and EBJIS criteria used in the present study. Moreover, the criteria were described in details in lines 134-135 and 139-151 with appropriate references.